# Investigation of Fiber–Matrix Interface Strength via Single-Fiber Pull-Out Test in 3D-Printed Thermoset Composites: A Simplified Methodology

**DOI:** 10.3390/ma17102433

**Published:** 2024-05-18

**Authors:** Kaan Nuhoglu, Neyton M. Baltodano, Emrah Celik

**Affiliations:** Mechanical and Aerospace Engineering Department, University of Miami, Coral Gables, FL 33146, USA; kxn390@miami.edu (K.N.); nxb707@miami.edu (N.M.B.J.)

**Keywords:** single-fiber pull-out, additive manufacturing, direct ink writing, single-fiber test

## Abstract

The emergence of additive manufacturing technologies for fiber-reinforced thermoset composites has greatly bolstered their utilization, particularly within the aerospace industry. However, the ability to precisely measure the interface strength between the fiber and thermoset matrix in additively manufactured composites has been constrained by the cumbersome nature of single-fiber pull-out experiments and the need for costly instrumentation. This study aims to introduce a novel methodology for conducting single-fiber pull-out tests aimed at quantifying interface shear strength in additively manufactured thermoset composites. Our findings substantiate the viability of this approach, showcasing successful fiber embedding within composite test specimens and precise characterization of fiber pull-out strength using a conventional mechanical testing system. The test outcome revealed an average interfacial strength value of 2.4 MPa between carbon fiber and the thermoset epoxy matrix, aligning with similar studies in the existing literature. The outcome of this study offers an affordable and versatile test methodology to revolutionize composite material fabrication for superior mechanical performance.

## 1. Introduction

Additive manufacturing of thermoset composite materials has emerged as a promising technology, owing to the recently developed extrusion-based Direct Ink Writing (DIW) process [1,2,3]. DIW of carbon fiber-reinforced polymer (CFRP) thermoset composites utilizes a viscous ink comprising thermoset resin and short-fiber reinforcement as primary components [4]. Moreover, this method facilitates the alignment of fibers while printing by utilizing the shear forces generated between the ink and the walls of the nozzle. Therefore, along with the benefits provided by 3D thermoset composite printing, the ability of the DIW process to customize fiber orientation according to specific application needs marks this method as a superior option for high-performance engineering applications [5,6,7].

The mechanical performance of fiber-reinforced polymer (FRP) composite structures is influenced not only by macro-level material properties such as volume fraction or fiber orientation but also by microstructure, as is widely acknowledged [8,9,10,11,12,13,14,15]. During the manufacturing of composite structures with additive manufacturing (AM) techniques, it is imperative to examine micromechanical properties to avoid deficiencies compared to traditional manufacturing techniques [16,17]. One of the crucial microstructural characteristics in FRP materials is the bonding strength formed between the fibers and the matrix material. To evaluate interfacial bonding strength, the fiber pull-out test is one of the most suitable and widely studied experimental methods in the literature [18,19,20]. During this test, interfacial shear stresses develop between the fiber and matrix, peaking as the fiber embeds into the matrix and gradually decreasing. Eventually, debonding occurs when the applied stress surpasses the maximum interfacial shear strength between adjacent surfaces of the fiber and matrix [21,22].

Although there are numerous studies characterizing the mechanical performance of additively manufactured thermoset composites, these focused on only the overall mechanical performance of these materials without consideration of the fiber–matrix interfacial strength via the single-fiber pull-out test [23,24,25]. The lack of scientific investigation in this area clearly demands further research to minimize the knowledge gap on the interfacial bonding strength of these composite materials. The complexity and high cost of single-fiber pull-out experiments can explain the limited knowledge in this area. Most studies in the literature focus mainly on numerical and analytical investigation of the fiber pull-out mechanism, with limited attention given to the improvement in the experimental test methodologies [23,24,25,26,27,28,29,30,31,32]. The earliest single-fiber pull-out experimental studies were conducted by M. R. Piggott et al. [33], E. Mäder et al. [34], and M.J Pitkethly et al., [35]. These researchers created assorted setups and apparatuses for both manufacturing and testing processes of both glass- and carbon fiber-embedded thermoset resins. The most prosperous study carried out in recent years was conducted by C. Kahl et al. [36], who investigated the matrix–fiber bonding mechanism employing advanced equipment, the Favimat+ (Moenchengladbach, Germany), which was recently introduced. On the other hand, A. Becker-Staines et al. [37], P. Chindaprasirt et al. [38], and E. Wolfer et al. [39] developed custom testing and manufacturing methodologies in the case of not only thermoset but also thermoplastic and concrete matrix materials. Despite the significance of these studies, there is a necessity to simplify the current methods for manufacturing and testing fiber pull-out test specimens. This simplification would facilitate widespread adoption without the requirement for costly setups or time-consuming procedures. In addition, there is a need for a novel methodology to perform these tests on additively manufactured composites where the composite microarchitecture and the fiber alignment can be precisely controlled.

Acknowledging the gap in the existing literature, this study aimed to elucidate the microscale interactions between fibers and the composite matrix in additively manufactured CFRP composites using the DIW technique. The research involved a collaborative effort between additive manufacturing and mechanical testing methodologies to accurately and simply characterize the interfacial strength between the thermoset matrix and individual fibers. The methodology outlined in this study is robust, enabling single-fiber pull-out tests across composites with varying composite microarchitectures. Moreover, it is adaptable to different fiber and matrix materials and does not necessitate expensive equipment, as testing can be conducted using standard mechanical testing instruments. The development of such a versatile single-fiber pull-out testing approach has the potential to enhance understanding of micromechanical properties in additively manufactured composites and bridge the existing knowledge gap in this area.

## 2. Materials and Methods

### 2.1. Composite Ink Preparation

The composite ink used in this study was composed of thermoset epoxy resin (EPON Resin 826, Hexion Inc., Columbus, OH, USA), chopped carbon fibers (Hexcel AS4/BR102), nanoclay (Garamite-7305 from BYK additives), and a latent curing agent (1-Ethyl-3-methylimidazolium dicyanamide, Sigma-Aldrich, Burlington, MA, USA). The chopped carbon fibers were used as a reinforcement material in the composite ink, while the nanoclay served as a rheological modifier to improve the self-supporting properties of the ink material. Nanoclay tremendously increases the viscosity of the printing ink, which allows for shape retention of the composite ink after the extrusion. Highly complex shapes can, therefore, be printed with highly viscous printing inks showing shear-thinning behavior. The individual fiber filaments measured 6.4 mm in length and had a diameter of 7.1 μm [40].

Each composite ink was prepared by adding 7 pph of nanoclay to epoxy resin followed by mixing in a Thinky ARE-310 high-shear planetary mixer. Carbon fiber was then added gradually to achieve 10 pph, followed by two two-minute mixing sessions at 2000 rpm for each addition. The ink was cooled down to ambient temperature before a 5 pph hardener was added to prevent premature curing. It should be noted that the percentages are given by parts per hundred (pph) by weight of the epoxy resin. After adding the hardener, the ink was then mixed under vacuum in three 30 s sessions at 1600 rpm to minimize air bubbles before being filled into a 30 cc syringe barrel.

### 2.2. Velocity Ratio and Rapid Fiber Alignment Analysis

The velocity ratio was utilized as the single parameter to dynamically control the fiber alignment of 3D-printed carbon fiber composites. Velocity ratio is a nondimensional speed metric that is the ratio of the composite exit velocity and the printing velocity. Therefore, if the velocity ratio is equal to 1, the material exiting the nozzle has the same velocity as that of the nozzle travel resulting in the nozzle diameter and the road width of the printed ink being nearly the same. If the velocity ratio is less than 1, then the composite is over-extruded leading to randomized fiber orientation. If the velocity ratio is greater than 1, then under-extrusion takes place narrowing the width of the printed road and increasing the chance of defect formation. In this study, a velocity ratio of 1 was selected to fabricate the composite specimens. To achieve this velocity ratio, the material exit velocity was first calculated by extruding a composite ink at a constant pressure for 10 s and measuring the amount of material that was extruded. Multiple measurements were taken to ensure consistency. Material exit velocity (*V*) was then calculated by using Equation (1), where m is the mass of the extruded material, ρ is the composite ink’s density, and d is the nozzle diameter.
(1)$$V=\frac{2m}{5\pi \rho d^{2}}$$

To quantify the fiber alignment in 3D-printed composites, the rapid fiber alignment analysis (RFAA) technique was utilized [6]. Single-road carbon fiber composite samples were initially 3D-printed onto a glass slide and optical microscope images of the composite ink were then captured using a Keyence VHX-5000 microscope positioned at the bottom side in contact with the glass slide. Subsequently, microscope images depicting fiber orientation were cropped to isolate relevant areas and converted to 8 bit grayscale using the image analysis software Fiji. The Ridge Detection plugin [41] in Fiji facilitated the extraction of individual fibers, generating a binary image. Finally, the FibrilTool plugin [42] was employed to quantify fiber alignment within the binary image. FibrilTool provides the fiber alignment score, a quantifying metric ranging from 0 to 1. A score of 0 denotes random fiber alignment, while a score of 1 indicates perfectly aligned fibers.

### 2.3. Placement of a Single-Fiber within 3D-Printed Composites

Single-fiber pull-out specimen fabrication started with printing half-tall composite specimens using a custom Direct Ink Writing (DIW) printing system. The square CAD model (13 mm × 13 mm) was loaded into the slicing (Slic3r 1.3.0) software to generate a g-code file with a total of 6 layers in the z-direction, using a layer height of 0.58 mm. The road width was set to 0.5 mm for all samples. All composite specimens were fabricated with a customized DIW system that employs a Creality Ender 5 FDM printer, an Ultimus V pressure regulator, and a California Air Tools air compressor (Figure 1A). The DIW system was equipped with a custom print head capable of housing an ink-filled 30cc barrel during the 3D printing process. Single-road samples were printed onto glass slides to assess the fiber orientation. For multi-layered samples prepared for fiber pull-out experiments, an aluminum substrate with milled slots was utilized as shown in Figure 1A. The substrate was attached to the printer’s build plate and was used to properly orient the glass slides. Glass slides were then inserted into these slots and the samples were printed with a 580 μm tapered nozzle on the glass slides. Before placing individual fibers, half-tall (3-layer) specimens were printed as shown in Figure 1A.

To separate individual fibers, ¼” chopped carbon fiber tow was placed in a Petri dish containing isopropanol solution. Under an optical microscope, the fibers were agitated gently with the tip of the tweezers. Once separated, individual fibers were moved out of the dish by the tweezers and placed on a glass slide. To minimize the fiber motion, a drop of epoxy with 1 pph nanoclay (without curing agent) was placed at the end of the slide (Figure 1B). Under the optical microscope, the fiber alignment was corrected and the overhang from the glass slide was kept at approximately 2.5 mm to be able to reach the printed composite specimen. The optical microscope was also used to measure the overall length of the individual fiber. The glass slide with the single fiber was then positioned in the slot of the aluminum substrate, near the half-printed composite specimen and gradually tilted so that the individual fiber could be placed at the center of the composite sample. The height of the 3rd layer was the same as the glass slide to avoid any out-of-plane misalignment in the z-direction. The printing process was then continued, and the remaining 3 layers were printed to complete the composite specimen containing a single fiber (Figure 1C). The completed 6-layered composite samples were cured in an oven at 100 °C for 15 h before mechanical testing. As the final step, the glass slides were removed after curing was completed. Since there was no curing agent in the epoxy drop (only epoxy with nanoclay) placed on the glass slide, the fiber could be removed from the glass slide easily without damaging the fiber. Prior to the pull-out experiments, the fiber alignment was checked under the optical microscope and the samples with misalignment in all directions exceeding 30° were discarded to minimize the error in the fiber pull-out experiments (Figure 1D). The free length of the embedded individual fiber was also measured using the optical microscope. This allowed us to determine the embedded length of the fiber since all fibers had their overall lengths measured prior to being embedded in the composite. In order to provide a better understanding of fiber-embedded lengths and angles, Figure 2 shows the average fiber-embedded length and angle measurements. As can be seen in these graphs, while the fiber angle achieved was 85 degrees on average without observing excessive deviation (Figure 2A), the average fiber-embedded value was obtained as 1646 µm with a relatively high deviation (Figure 2B), which may have a significant influence on the test results.

Energy Dispersive Spectroscopy (EDS) analysis using the JOEL JSM-6010 PLUS/LA Scanning Electron Microscope (SEM) was conducted on the pull-out specimen and the reference carbon fiber to explore the contamination on the fiber surface during specimen manufacturing. The analysis was performed on 3 different locations on both fibers. Appendix A shows the SEM pictures of the specimen and reference fibers mounted on a carbon tape substrate. The figure shows the spots on the fibers from which we took measurements: the red spots are the locations we analyzed on the fiber pull-out specimen, while the blue spots are the ones measured on the reference fibers. Appendix A shows the average mass % values for the real specimen and reference fiber. As a result, 3 common elements (C, O, and N) were identified for both fibers, and the average mass % values for each element were remarkably compatible between real and reference fibers. The comparison validates that there was no change in the elemental composition and the fibers were not contaminated during the manufacturing process. In other words, the inserted fiber has the same adhesion property as the short carbon fibers in the paste ink.

### 2.4. Fiber Pull-out Test Procedure

Prior to fiber pull-out testing, the glass slide containing the composite specimen with a single fiber was attached to the Instron 5564 universal testing system (Instron, Norwood, MA, USA), as depicted in Figure 3A. To fasten the free end of the fiber, a disposable fixture 3D-printed using ABS material was placed on the bottom of the test frame. This fixture was disposed of after each test and replaced with another one to expedite the testing and cleaning process in these experiments. A strong epoxy adhesive (J-B Weld Steel) was placed in the hole at the center of this fixture, as seen in Figure 3B. A set screw was used to attach the fixture to the test frame to prevent any motion during testing. With all the Instron fixtures in place, the test specimen was lowered slowly until the individual fiber was embedded about 2 mm into the epoxy. The test setup was then left undisturbed for a minimum of 20 h to achieve the complete curing of the J-B Weld epoxy around the fiber.

Single-fiber pull-out tests were carried out under laboratory conditions (2 ± 2 °C at 50% ± 5 relative humidity) using Instron 5564 test equipment equipped with a 100 N static load cell. The displacement-controlled tensile test was performed under quasistatic crosshead speed rates (0.1–0.5 mm/min). The load–displacement (*P*-*δ*) data were recorded with the data acquisition system embedded in the Instron. After acquiring the load–displacement data, the interfacial shear stress (τ_max_) values were calculated using Equation (2), as suggested by C. DiFrancia et al. [28], by taking the diameter of the fiber (*d*) and its embedded length in composite (*l*) to determine the area of the fiber in contact with the composite matrix.
(2)$$\tau_{max}=\frac{P}{\pi dl}$$

The embedded length was calculated by subtracting the fiber-free length from the overall fiber length. To ensure that the fiber pull-out force is less than the fiber pull-out from the J-B Weld epoxy at the bottom fixture, the length of the fiber within the composite was designed to be at least 1 mm shorter compared to that within the epoxy fixture. The longer embedded length within the fixture led to stronger adhesion between the fiber and the J-B Weld adhesive and minimized the risk of fiber pull-out from the bottom test fixture. To prevent slipping, all specimens and test components were tightly attached and/or glued during mechanical testing. However, it may still be possible to observe some slippage during testing due to manual clamping procedures. In this paper, however, the main conclusions were drawn on the strength data, which are not affected by the slippages.

## 3. Results and Discussion

### 3.1. Fiber Alignment within the Composite

DIW allows shear alignment of the fibers by controlling the velocity ratio during printing. In this study, the velocity ratio was kept to 1 in order to equal the extrusion and printing speeds. To accurately determine the interfacial strength between the single fiber and the composite matrix, fiber alignment within the composites should be similar for each ink prepared for printing. Figure 4A shows an example of the single road specimen to be used to calculate the fiber alignment score. As seen in this figure, fibers are well aligned in the printing direction. Some misalignment was also observed, especially around the air bubbles within the composite specimen. Figure 4B shows the fiber alignment scores for four different inks used to prepare eight specimens for the pull-out tests. Although these inks were prepared on different days, similar fiber alignment scores were obtained for all inks, which eliminated the effect of fiber alignment within the composite pull-out tests for these specimens. Figure 4C shows the alignment of the single fibers with respect to the fiber pull-out test directions. The specimen on the left shows nearly perfect alignment of the single fiber. The middle specimen shows a light misalignment (less than 30°) with respect to the pull-out direction. Despite its misalignment, this specimen was included in the test data. However, the last specimen on the right had a misalignment angle exceeding the allowed 30° and, therefore, it was discarded and not used in the mechanical test results presented in the next section.

A detailed SEM analysis was conducted to provide a better understanding of the microstructure of the specimens fabricated. Appendix A presents the SEM image of the pull-out specimen before (Appendix A) and after (Appendix A) the tensile testing experiment was conducted. Appendix A also provides the microstructural details of the pull-out specimens. Since the specimens were produced via 3D printing technology, justifiably, concerns about potential porosity or weaker interface properties may arise. To address those concerns, Appendix A shows that there is no visible porosity between printing layers, validating that adhesion between the fiber and matrix interface is strong during mechanical testing. Since the material is in paste form, the bottom and top layers of the printed ink wet the surface of the inserted fiber, and a strong connection is obtained.

### 3.2. Fiber Pull-out Test Results

Figure 5 represents the load–displacement curves of the fiber pull-out tests performed on 14 test specimens. Since the test involves a debonding process, nonlinear behaviors can be seen for all specimens regardless of whether a complete or semi-pull-out was observed. Similarly, in all samples, when the maximum force was reached after the loading zone, the fracture was observed as a sudden load drop. This figure denotes that the fiber pull-out mechanism shows significant variance, which could be due to the imperfections in the fabricated specimens. As stated in the previous section, not all fibers have ideal alignment angles, and these specimens consist of significant levels of air bubbles, which may affect the interfacial adhesion strength between the fiber and the composite matrix. Out of 14 specimens tested, complete fiber pull-out was observed on 4 specimens. However, in the remaining samples, the fibers broke off after the maximum load was achieved due to the aforementioned imperfections and semi-pull-out was observed in these specimens. The pull-out behavior was confirmed by observing the samples under a microscope after the test. In this regard, while complete pull-out was observed in specimens 3, 5, 6, and 7, the test resulted in semi-pull-out in the rest of the samples.

Another observation from Figure 5 was the bimodal fiber pull-out behavior with small (<0.06 mm) and large (>0.06 mm) pull-out displacements. Extended displacements with slower stiffness could be the result of sliding or pull-out from the bottom adhesive in addition to the fiber pull-out from the composite. As explained in Section 2.3, the fiber-embedded length in the bottom fixture was kept longer than that of the composite to minimize the fiber pull-out from the J-B Weld adhesive from the fixture. However, these embedded lengths showed variation due to the manual labor in this scale, which may cause some slipping or pull-out from the J-B Weld.

Figure 6A shows the interfacial fiber pull-out strengths of the 14 tested specimens. Equation (1) was employed to calculate the interfacial shear strength between the fiber and the epoxy matrix. As can be seen in the test results, the interfacial shear stress values on the fiber pull-out specimens varied in the range of 1.4 MPa and 5.1 MPa. Figure 6B shows the average interfacial shear strength values obtained as a result of the fiber pull-out test performed on the DIW 3D-printed carbon–epoxy composite specimens. The average interfacial strength value was calculated as 2.45 MPa with a 1.15 standard deviation. The averaged interfacial fiber pull-out strength (2.45 MPa) was in agreement with the previously published results for similar composite material systems. The average result obtained is convergent when the studies in the literature are taken into consideration [43,44]. As explained previously, imperfections within the composite may lead to a large standard deviation in these test data.

## 4. Conclusions

The single-fiber pull-out test is the main testing procedure to quantify fiber–matrix interface strength. With the advent of thermoset composite additive manufacturing via the DIW method, there is a need to quantify the fiber adhesion strength within these composite systems. This study investigated a novel approach to preparing single fibers placed within the center of 3D-printed composite structures and pulling these fibers using a tensile testing setup. Unlike the other studies in the literature, this approach allowed interfacial strength of individual fibers protruding from the fiber-reinforced composite where the fiber alignment could be controlled with the velocity ratio. In addition to its versatility, this approach did not require any specialized testing and characterization instrument and all experimental procedures could be performed in a standard material testing laboratory setting.

The results have shown that the composite inks prepared on different days displayed similar fiber alignment behavior, as quantified by the fiber alignment scores from the single road images. Fiber pull-out specimens showed physical imperfections and defects, such as different embedded fiber lengths, fiber alignment, and porosity. As a result, fiber pull-out test results exhibited significant variation.

Despite its novelty, simplicity, and versatility, the developed single-fiber test procedure could be further improved. Minimizing the fiber alignment would further enhance the accuracy of the test setup. In addition, porosity could be lowered by vacuum mixing the composite ink to further minimize the test variability. To avoid the fiber pull-out from the bottom fixture and ensure that fiber is pulled from the composite system only, longer fibers could be utilized and the amount of embedded length in the J-B Weld within the bottom test fixture could be further extended.

## Figures and Tables

**Figure 1 materials-17-02433-f001:**
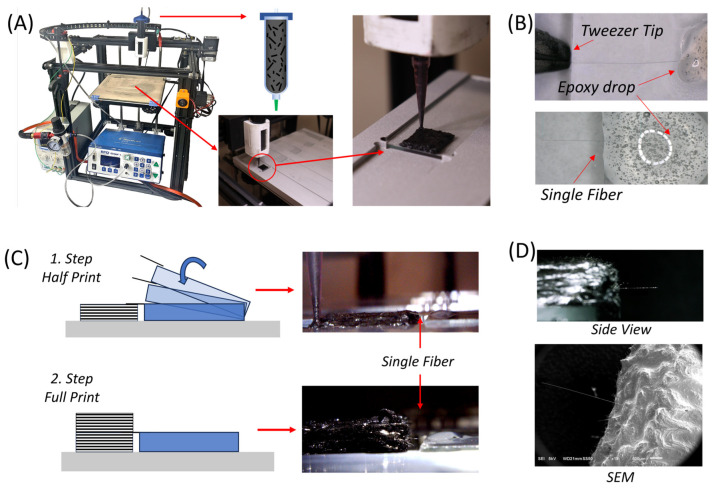
DIW 3D printing system utilized to fabricate fiber pull-out specimens. (**A**) DIW printer and redesigned build plate. (**B**) Placing the single fiber into an epoxy drop. (**C**) Specimen manufacturing steps. (**D**) Single-fiber pull-out specimen side view and SEM pictures.

**Figure 2 materials-17-02433-f002:**
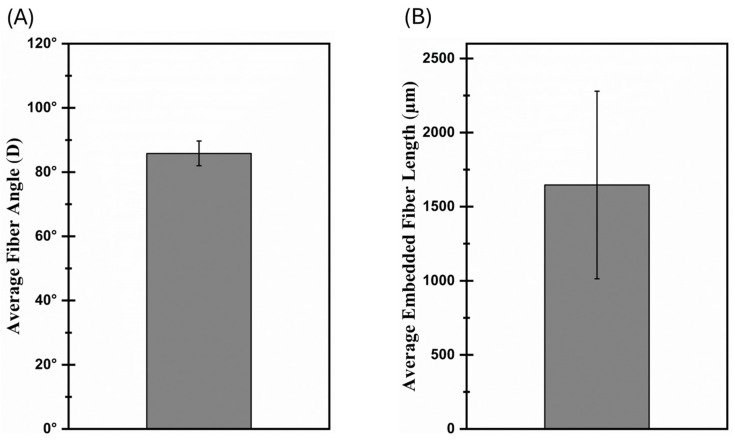
Imperfections in fabricated samples: (**A**) fiber orientation angle, (**B**) embedded fiber length.

**Figure 3 materials-17-02433-f003:**
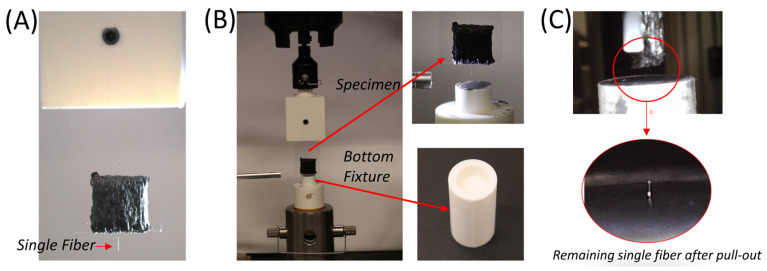
(**A**) Composite sample attached to the mechanical test frame, (**B**) bottom fixture attached to Instron and insertion of the single fiber, and (**C**) completed pull-out test and separation of the fiber.

**Figure 4 materials-17-02433-f004:**
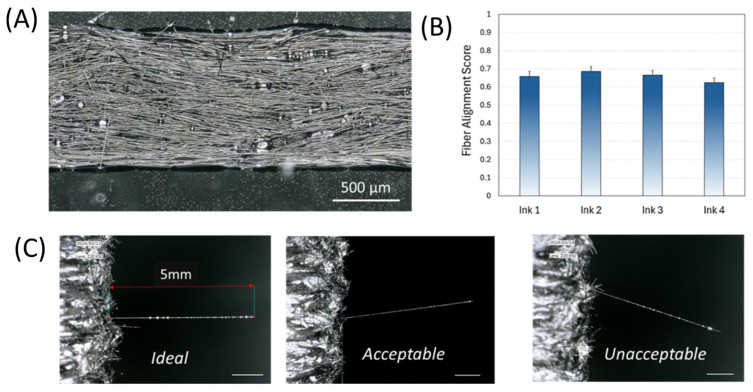
(**A**) Optical microscope image of a single road sample, (**B**) fiber alignment score graph, and (**C**) examples of ideal, acceptable, and unacceptable fiber angles.

**Figure 5 materials-17-02433-f005:**
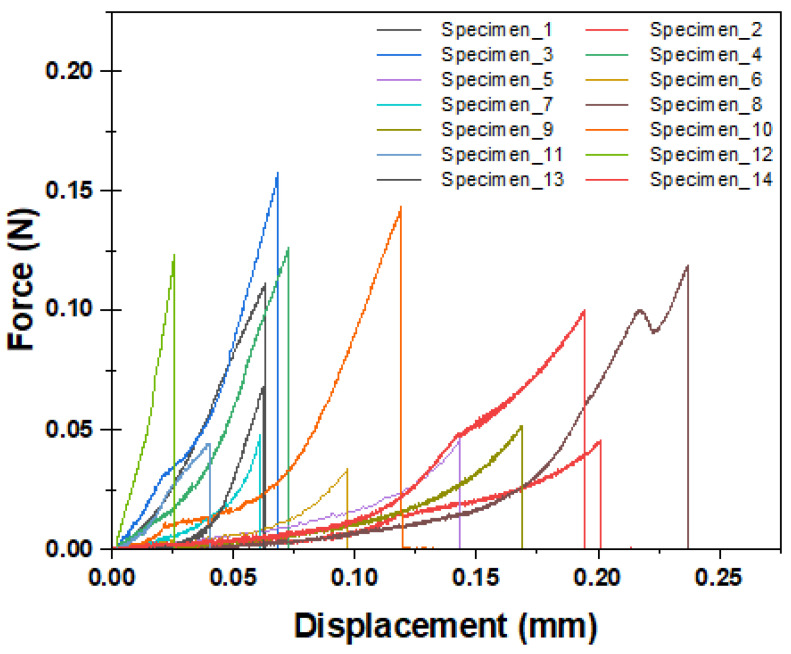
Load–displacement graph of the single-fiber pull-out specimens.

**Figure 6 materials-17-02433-f006:**
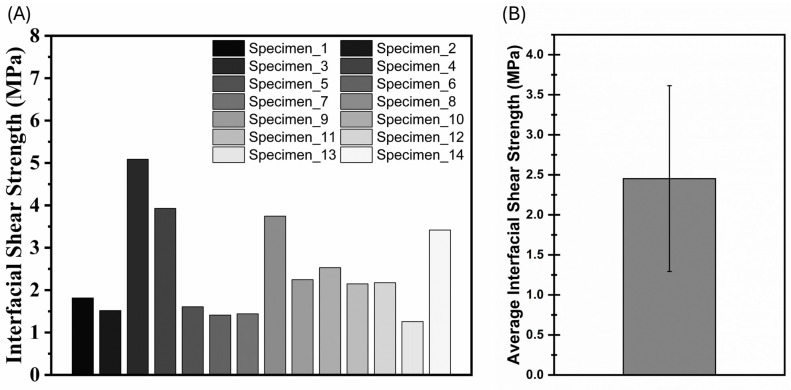
(**A**) Interfacial shear strength graph of the single-fiber pull-out specimens. (**B**) Average interfacial shear strength graph.

## Data Availability

Data collected in this study are available upon request.

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
