# Peer review of "Investigation of Fiber–Matrix Interface Strength via Single-Fiber Pull-Out Test in 3D-Printed Thermoset Composites: A Simplified Methodology"

_materials, 2024, doi:10.3390/ma17102433_

Round 1

Reviewer 1 Report

Comments and Suggestions for Authors

Please, see attached file.

Reviewer 2 Report

Comments and Suggestions for Authors

The paper is very good work, but I have two requests before its acceptance.

First, please, provide more detailed SEM observations of the morphology.

Second, to provide spectroscopic analysis to verify the contamination of the fiber surface by matrix residues.

Reviewer 3 Report

Comments and Suggestions for Authors

his manuscript studied the problem and strength of pull-out phenomenon in 3-D Printed Thermoset Composites. Interesting paper, however my comments are as follows,

 1.      5pph, 30s, 1600rpm, 30cc, 580μm; please leave a space between value and unit.

2.      Line 37: What is AM?

3.      What is observed44 on line 231?

4.      τmax on line 193

5.       It seems that there are 2 trends of the load-displacement results, one group are curves 2, 5, and 6 (ductile behavior, might cause by slip as well) and another are curves 1, 3, and 4 from Figure 4. Why the load on curve 7 is low? Maybe 7 samples are not good enough for sampling.

Reviewer 4 Report

Comments and Suggestions for Authors

I agree with the authors on the expense of single-fibre pull-out test equipment. Thus, any procedure to obtain the value for the interfacial shear strength is welcome.

I do not know the use of thermosets in FDM so one of my doubts is how to control the cure of the resin without compromising the shape of the model.

Figure 1, the top view of the fiber. It is difficult to see the fiber, please add some marks (circle, ellipse, arrow) to make the identification of such fiber.

I see a large scatter of the results and two different mechanisms in figure4. The fibers with a larger displacement can have some kind of slippage and the others with higher Force and less displacement are under pure tensile. Usually we test five specimens, but in this case, I think that the number of test specimens must be higher, and also a strategy to identify non-standard fiber behavior can be useful. I ask the authors for their opinion on the subject.

Another question is how to measure the displacement. There are a lot of materials between the clamps and the displacement can be overestimated.

Round 2

Reviewer 1 Report

Comments and Suggestions for Authors

The reviewer would like to acknowledge the level of detail in the authors' responses. Not only have they justified and corrected what was requested, but they have also doubled the number of tests carried out, and have included new microscopic (SEM) and chemical characterizations. The new results included correctly justify the conclusions, and answer all the reviewer's doubts, so the article is in a position to be accepted for publication. Congratulations on the work.